# S-MAT: Semantic-Driven Masked Attention Transformer for Multi-Label Aerial Image Classification

**DOI:** 10.3390/s22145433

**Published:** 2022-07-20

**Authors:** Hongjun Wu, Cheng Xu, Hongzhe Liu

**Affiliations:** 1Beijing Key Laboratory of Information Service Engineering, Beijing Union University, Beijing 100101, China; gwawhj@126.com (H.W.); xc-f4@163.com (C.X.); 2Institute for Brain and Cognitive Sciences, Beijing Union University, Beijing 100101, China

**Keywords:** aerial scene classification, multi-label learning, redundancy removing, label correlation, semantic disentanglement

## Abstract

Multi-label aerial scene image classification is a long-standing and challenging research problem in the remote sensing field. As land cover objects usually co-exist in an aerial scene image, modeling label dependencies is a compelling approach to improve the performance. Previous methods generally directly model the label dependencies among all the categories in the target dataset. However, most of the semantic features extracted from an image are relevant to the existing objects, making the dependencies among the nonexistant categories unable to be effectively evaluated. These redundant label dependencies may bring noise and further decrease the performance of classification. To solve this problem, we propose S-MAT, a Semantic-driven Masked Attention Transformer for multi-label aerial scene image classification. S-MAT adopts a Masked Attention Transformer (MAT) to capture the correlations among the label embeddings constructed by a Semantic Disentanglement Module (SDM). Moreover, the proposed masked attention in MAT can filter out the redundant dependencies and enhance the robustness of the model. As a result, the proposed method can explicitly and accurately capture the label dependencies. Therefore, our method achieves CF1s of 89.21%, 90.90%, and 88.31% on three multi-label aerial scene image classification benchmark datasets: UC-Merced Multi-label, AID Multi-label, and MLRSNet, respectively. In addition, extensive ablation studies and empirical analysis are provided to demonstrate the effectiveness of the essential components of our method under different factors.

## 1. Introduction

Recently, the great progress in remote sensing technologies has provided increasing remote sensing images from satellite-borne and airborne sensors for land cover mapping and monitoring issues. Generally, since high-resolution remote sensing images depict diverse categories of land cover objects, a single label cannot accurately describe the content in the image. Therefore, compared with single-label classification [1,2,3], multi-label remote sensing image classification is a more practical task. Specifically, a Multi-Label Image Classification (MLIC) method is developed to assign a set of preset land cover labels to each remote sensing image. In this paper, we focus on the multi-label remote sensing image classification task in the aerial scene understanding field.

Benefiting from the great success of deep learning, deep Convolutional Neural Networks (CNNs) [4,5] and vision Transformers [6,7] are proposed to extract high-level semantic features and make incremental progress in Single-Label multi-class Image Classification (SLIC). Additionally, the field of numerical simulation and stability has achieved significant progress [8]. These advanced methods are also exploited in single-label remote sensing image classification. By treating each label in isolation, the multi-label problem can be simply addressed by using SLIC methods to predict whether each label is present or not. However, compared to SLIC, MLIC is a more complicated task. On the one hand, in an aerial image, there are multiple land cover objects at different spatial resolutions, which are related to the size of the objects. For example, the size of a car is far less than a court, and consequently, “car” is one of the inconspicuous categories. On the other hand, since land cover objects generally co-exist in an aerial scene image, the inter-class relationship is another key for the classification. Therefore, the MLIC task considers not only accurate spatial feature extraction, but also the correlations of multiple concepts. In classical MLIC, the utilization of spatial information and inter-class correlations are both significant issues. To handle the spatial information, some works introduce regional proposal techniques [9,10], implicit spatial attention [11,12], or multi-scale features [13]. Nevertheless, these methods neglect the impact of the relationships among multiple categories. On the other, many works are proposed to model the inter-class correlations. Pioneering approaches [14,15,16,17,18] based on Recurrent Neural Networks (RNNs) or Long Short-Term Memory (LSTM) learn the correlations in sequential prediction. However, the performance of the RNN-based methods is influenced by the pre-set or learned sequence. Moreover, the complex label correlations cannot be accurately represented by sequential relations. Other works [19,20] formulate the MLIC task as a structural inference problem based on probabilistic graphical models [21], while the practicality is limited by the high computational complexity. Inspired by the great success of the Graph Convolutional Network (GCN) [22] in the representation of multivariate relations, ML-GCN [23] proposes to explicitly model label correlations via GCN. Transformer [24] from the Natural Language Processing (NLP) field has achieved great success [25,26,27] in the Computer Vision (CV) area. Inspired by the impressive capability to model long-range dependencies, recent works [28,29] leverage Transformer to capture the label dependencies.

Previous methods have proven the effectiveness of dealing with label dependencies. In general, these methods directly model the holistic label dependencies among the generated label embeddings related to all categories. Nevertheless, only a portion of category objects is present in a single image, and the visual features extracted from the image are mostly relevant to the ground-truth labels. Consequently, the computed label dependencies among the nonexistent categories are inaccurate. In this paper, these label dependencies are called redundant dependencies, which bring noise to the classification task. Specifically, as shown in Figure 1, the solid line indicates a higher connection and the dashed line indicates a lower connection between categories. According to the presence or absence of the categories in the image, some of the inter-class relationships are obvious and definite, such as the solid lines (higher connections) and dashed grey lines (lower connections). However, dependencies among nonexistent categories (e.g., car and court) cannot be accurately estimated, which leads to a negative impact on classification.

In short, there are mainly three challenges in multi-label remote sensing image classification tasks. Firstly, land cover objects have different scales in an image, making it more challenging to leverage spatial information. Secondly, there are usually co-occurrence dependencies among the objects of different categories in a remote sensing image. Thirdly, the redundant correlations among the nonexistent categories bring noise to multi-label remote sensing image classification.

To this end, we propose a novel and effective multi-label image classification framework, Semantic-driven Masked Attention Transformer (S-MAT), which consists of a backbone feature extractor, a Semantic Disentanglement Module (SDM), and a Masked Attention Transformer (MAT). In this paper, we first extract the high-level semantic feature of a remote sensing image with a CNN backbone. Then, the extracted feature is disentangled into a set of label embeddings in the SDM. After that, a Masked Attention Transformer (MAT) is trained to model label correlations and update the input label embeddings adaptively. Meanwhile, we exploit the masked attention in MAT to restrict the attention to the categories with higher confidence and avoid the noise from nonexistent categories. Finally, the updated label embeddings are projected to the image-level predictions, which are combined with the independent predictions generated directly from the high-level image representation to obtain the final predictions. In addition, we notice that masked attention is applied to other visual tasks, such as Mask2Former [30] for image segmentation. The attention mask in Mask2Former is class agnostic, whereas, in our work, each element $M_{ij}$ in attention mask *M* corresponds to the relevance between class *i* and *j* in the attention map.

Comprehensive results on three widely used multi-label image recognition benchmarks show that our S-MAT outperforms other recent methods that model label relationships via graph convolution networks or other proposed strategies. In summary, our main contributions are as follows:A novel Transformer-based framework, S-MAT, namely Semantic-driven Masked Attention Transformer, is proposed. S-MAT aims to filter out the redundant dependencies and obtain more accurate label dependencies for multi-label aerial scene image classification.We conduct in-depth studies on the application of masked attention and propose a plug-and-play module, Masked Attention Transformer (MAT), to constrain the attention to the categories with higher confidence and reduce the redundant dependencies among the nonexist classes. To our best knowledge, this is the first application of masked attention in modeling inter-class relationships.We design a plug-and-play module, namely the Semantic Disentanglement Module (SDM), to disentangle the high-level semantic feature into a set of category-relevant embeddings for each image by locating the attention region of each category.We conduct comprehensive experiments to verify the effectiveness of the proposed approach. On three widely used multi-label aerial scene image recognition benchmarks including UC-Merced Multi-label, AID Multi-label, and MLRSNET, our models consistently have state-of-the-art results.

The rest of this paper is structured as follows. Section 2 gives the related works. Section 3 demonstrates the details of the structure and relative setting of the proposed method. Section 4 is devoted to the discussion of the experiments. Section 5 gives the ablation studies. Section 6 gives the qualitative results. Section 7 analyzes the experimental results and discusses the difference among the proposed method and the previous methods. Section 8 presents the conclusion.

## 2. Related Work

Multi-label aerial scene image classification plays a vital role in the imagery interpretation for remote sensing images. Recently, significant progress has been achieved in multi-label image classification tasks. The proposed approaches for multi-label image classification can be roughly categorized into two aspects, i.e., spatial information and label correlations.

### 2.1. Spatial Information

The utilization of spatial information is key to improving the performance of visual recognition, especially multi-label image recognition. The objects in different locations in an image are usually present at different scales. Pioneering works [9,10] introduce the regional proposal technique from the object detection field and transfer the task into the SLIC tasks in each generated proposal. However, accurate regional proposals need the supervision of extra object-level annotations, which are much more costly than image-level annotations. Hence, some approaches replace the explicit proposals with semantically relevant attentional regions. Wang et al. [11] propose using a spatial Transformer network [31] to ascertain the interest regions corresponding to the semantic labels and predict the scores via Long Short-Term Memory (LSTM) [32]. MCAR [12] presents a two-stream network to recognize multi-category objects from a global image to local regions and designs a multi-class attentional region module to generate a smaller number of attentional regions. Xiong et al. [33] propose a Confounder-Free Fusion Network (CFF-NET) to extract fine-grained deep features from the whole image and provide more multi-grained image information via visual attention. Liang et al. [13] extract multi-scale image features by using multi-scale Graph Convolutional Networks (GCNs).

### 2.2. Label Correlations

Since the co-occurrence of objects in images conforms to the general rules of the real world, mining the label distribution as prior knowledge of subsequent classification enhances the performance of multi-label image classification.

#### 2.2.1. RNN-Based Methods

The CNN-RNN [14] framework learns label correlations with an LSTM layer. Order-free RNN [15] introduces confidence-ranked LSTM and visual attention to increase the flexibility of the label sequence. Orderless RNN [16] proposes an orderless loss, which can dynamically order the labels based on the prediction of the LSTM model to reduce duplicate prediction. Hua et al. [34] exploit the class attention learning layer to generate category-specific features and use a bi-LSTM to model the label dependency in both directions and produce structured multiple object labels. Ji et al. [17] introduce the attention module to separate the high-level semantic features by channel and sequentially predict labels via an LSTM network. Wang et al. [18] propose a Semantic Supplementary Network with Prior Information (SSNP) to first generate prior information by using an LSTM-based prior information network and generate semantic information of the potential labels via a semantic supplementary module.

#### 2.2.2. Graph-Based Methods

Compared to the sequential methods, graph-based approaches have attracted more attention. Early works coped with such a dependency via probabilistic graph models, such as the cyclic directed graphical model [35] and tree-structured graph model [19]. The Graph Convolutional Network (GCN) [22] has been proven to be a more effective tool in modeling label correlations. Chen et al. [36] propose to construct a directed graph based on statistical label co-occurrence and word embedding of labels and propagate the inter-class information by the GCN. Besides the relationships among pairwise labels, Wu et al. [37] model the high-order semantic dependencies via hypergraph neural networks. The construction of the graph usually needs the pre-defined correlation matrix, which is set up with the statistic in the dataset (e.g., statistical label co-occurrence). The static graph cannot accurately represent the characteristic of an individual image. To tackle this problem, Ye et al. [38] use high-level spatial features to build an updatable graph via a Dynamic Graph Convolutional Network (D-GCN) module. In the remote sensing field, graph-based methods have been widely researched. Chaudhuri et al. [39] propose to build an image neighborhood graph via a semi-supervised graph-theoretic method for multi-label remote sensing image retrieval. Tan et al. [40] convert low-rank representation to a feature-based graph and a semantic-based graph, respectively. Similarly, Zhang et al. [41] build an image feature graph and a semantic graph. The difference is they regularize dual graphs via a non-negative matrix tri-factorization-based collaborative filtering framework. Li et al. [42] construct a scene graph with the extracted feature and mine the spatio-topological relationships of the scene graph via a Graph Neural Network (GNN). Li et al. [43] propose a CM-GM framework to align the image feature and label feature via a GCN, and then, the aligned features are fed into bi-LSTM to predict the image-level labels.

#### 2.2.3. Transformer-Based Methods

Transformer [24] is a conspicuous architecture that exploits self-attention to model positionwise relationships among the elements in a long sequence. Unlike CNNs or RNNs, Transformer demonstrates a greater capability for long-range modeling and adaptability to multiple domains, whether NLP or CV. Recently, Transformer has been explored in multi-label classification to model the label correlations [28,44]. Lanchantin et al. [44] propose to capture the dependencies among a set of label embeddings and adaptively combine spatial features via a Transformer encoder. Rather than label embeddings, Chen et al. [28] send category-specific activation maps into the Transformer encoder to exploit the relationships among categories. In the remote sensing field, Deng et al. [45] jointly train a CNN and a vision Transformer to combine the local and global features. Tan et al. [29] ratiocinate the inter-class relation matrix in a Transformer-based SRBM module to generate a robust semantic relationship category representation. Transtl [46] aims to adaptively locate the interested region of each label via one or more STLD modules.

Additionally, there have been other studies of multi-label recognition in the remote sensing field. Wang et al. [47] propose a Multi-Label Semantic Feature Fusion (MLSFF) framework, which consists of a multi-label semantic attribute extractor to extract multi-label semantic attributes and two cross-modal semantic feature fusion operators that fuse the extracted semantic attributes and the image feature extracted by the convolutional neural network. Yu et al. [48] propose a Self-Correction Integrated Domain Adaptation (SCIDA) method for automatic multilabel learning, including a Labelwise Self-Correction (LWC) module to better explore underlying label correlations.

## 3. Methods

In this section, we introduce a novel Multi-Label Image Classification (MLIC) framework named Semantic-driven Masked Attention Transformer (S-MAT), which provides a Transformer-based solution to make use of inter-class relationships to improve classification performance. This section consists of four parts. We first review Transformer in Section 3.1. Then, we introduce the Semantic Disentanglement Module (SDM) in Section 3.2 and the Masked Attention Transformer (MAT) in Section 3.3. In the end, we briefly describe the final classification and loss function in Section 3.4.

### 3.1. Recap of Transformer

The standard Transformer [24] architecture is a typical encoder–decoder architecture. This work is based on the Transformer encoder; thus, we shall introduce the Transformer encoder in the following. The Transformer encoder is a multi-level architecture, in which each layer comprises two key components, namely multi-head self-attention module and feed-forward network module. In the area of natural language processing, the conventional Transformers build relationships among different semantic words in the input language sentences from global perspectives. In other areas, the non-serialized input data need to be preprocessed into a sequence. For example, ViT [6] proposes to cut the image into multiple patches and flatten them into a sequence. Since the sequence loses positional information on the input, position embedding is introduced to preserve the relative position of each element in the sequence.

Given a sequence $X\in R^{l_{x}\times D}$, where $l_{x}$ is the length of *X*, they are converted into queries *Q*, keys *K*, and values *V* by the fully connected layers.
(1)$$Q=XW^{Q},K=XW^{K},V=XW^{V},$$
where $W^{Q}\in R^{D\times d^{k}}$, $W^{K}\in R^{D\times d^{k}}$, and $W^{V}\in R^{D\times d^{v}}$ are the learnable parameters for channel transformation. $d^{k}$ and $d^{v}$ are the channel numbers of the key and value. In this paper, we set $d^{k}=d^{v}=512$. The standard dot-product attention with the residual path is defined in Equation (Equation 1).
(2)$$A\left(Q,K,V\right)=\mathrm{Softmax}\left(\frac{QK^{T}}{\sqrt{d_{k}}}\right)V+Q.$$

Then, the updated query embedding *Q* is subjected to a fully connected Feed-Forward Network (FFN) to perform a nonlinear mapping. The FFN consists of two linear transformations with a GELU activation in between. The FFN with a residual path is as follows:(3)$$FFN\left(Q\right)=GELU\left(QW_{1}+b_{1}\right)W_{2}+b_{2}+Q,$$
where *W* and *b* stand for the weight matrices and the bias. The subscripts represent different fully connected layers. In this paper, we set the dimension of $W_{1}$, $W_{2}$, $b_{1}$, and $b_{2}$ as $W_{1}\in R^{512\times 2048}$, $W_{2}\in R^{2048\times 512}$, $b_{1}\in R^{2048}$, and $b_{2}\in R^{512}$, respectively.

In our approach, we replace the standard dot-product attention operator with a masked attention operator based on the meta-architecture mentioned above. The overview of our S-MAT framework is presented in Figure 2. It consists of four main parts: (i) high-level feature extraction of the input image via a pre-trained backbone, (ii) construction of label embeddings in te Semantic Disentanglement Module (SDM), (iii) relationship modeling and embedding refinement in a Masked Attention Transformer (MAT), and (iv) computing the final prediction logits for each category. Note that our method can be attached to any backbone without intrusive modifications. The detail will be introduced in the next part.

### 3.2. Semantic Disentanglement Module

Given an image $I\in R^{3\times H_{0}\times W_{0}}$, where $H_{0}$ and $W_{0}$ are the height and the width of *I*, high-level feature map $X\in R^{D^{\prime }\times H\times W}$ is extracted from the backbone. In this paper, the label embeddings are constructed by disentangling the image feature map *X*. The disentanglement is divided into two steps: (1) generation of class-specific activation *M*; (2) matrix product of *X* and *M*. Class-specific activation represents the probability of a label appearing at each spatial location, and its generation can be formulated as follows:(4)$$M=\sigma \left(\varphi_{m}\left(X\right)\right)\in R^{C\times H\times W},$$
where $\varphi_{m}\left(\cdot \right)$ denotes the 1×1 convolution layer to transform the dimension of *X* from $D^{\prime }$ to the number of classes *C* in the current dataset and σ(·) is the Sigmoid(·) function. In this paper, $D^{\prime }$ was set to 2048. Each element $m_{ij}^{n}$ in class-specific activation $M\in R^{C\times H\times W}$ represents the probability of specific category *c*’s presence in the feature map *X* at (i,j). We adopt Global Max Pooling (GMP) on *M* to generate the predictive logit $y_{a}=\left[y_{a}^{1},y_{a}^{2},\ldots ,y_{a}^{C}\right]$, which is constrained by the loss demonstrated in Section 3.4 for learning a more accurate representation *M*. Hence, the label embeddings are constructed by the product of *M* and the transformed feature map as follows:(5)$$E=R\left(M\right)\varphi_{c}\left(R{\left(X\right)}^{⊤}\right)\in R^{C\times D},$$
where $\varphi_{c}$ denotes the 1×1 convolution layer to reduce the dimension of *X* from $D^{\prime }=2048$ to D=512 and R represents the reshape operation, which squeezes the spatial dimensions *H* and *W* into one dimension HW. Intuitively, each category *c* selects the interested region in the transformed feature map to combine. As a consequence, each embedding $e_{c}$ aggregates the corresponding spatial feature and semantic information.

### 3.3. Masked Attention Transformer

Transformer has proven its outstanding ability to model long-range dependencies. In particular, the built-in mask matrix, which restricts the scope of attention, makes Transformer a perfect choice for modeling label relationships. The mask matrix in Transformer was originally intended to eliminate the effect of padding on the sequence in training or avoid exposing the decoder to predictive content in machine translation. In Mask2Former, the mask matrix is exploited to realize local attention by constraining the attention to the foreground region, instead of the full feature map. As mentioned in Section 1, one crucial problem in multi-label classification is removing the redundant part and obtaining more accurate label dependencies. To solve this problem, we make the first attempt to introduce masked attention into multi-label classification to mask the redundant label dependencies, and the attention is confined to the categories with higher confidence. The proposed masked attention is shown in Figure 3.

Suppose the ground-truth label set of the input image is $C_{1}$, which contains $n_{1}$ labels, and the set of nonexistent labels is $C_{2}$, which contains $n_{2}$ labels. D(X,Y) represents the relationship among the labels in both label sets X and Y. We believe that there are higher relations among the labels in C1, namely $D\left(C_{1},C_{1}\right)={\left\{1\right\}}_{n_{1}\times n_{1}}$. On the contrary, $D\left(C_{1},C_{2}\right)={\left\{0\right\}}_{n_{1}\times n_{2}}$. However, $D(C_{2},C_{2})$ is uncertain and redundant. If we filter out the redundant part, we can obtain more accurate label relationships. In this paper, we simply judge the confidence of each label according to $y_{a}$ and then generate the mask M. Since $y_{a}$ is not completely accurate, we retain some redundancy in the generation of masks by adjusting the proportion to be filtered out.

To generate the attention mask M, we first obtain *I*, the index set of the top-k prediction in $y_{a}$, via the topK operator in PyTorch. Then, the attention mask M is
(6)$$M\left(x,y\right)=\left\{\begin{array}{cc} -\infty & \mathrm{if}\;x,y\notin I\\ 1 & \mathrm{otherwise} \end{array}\right..$$

Thus, the masked attention $A_{m}$ can be defined as follows:(7)$$A_{m}=\mathrm{Softmax}\left(\frac{QK^{T}}{\sqrt{d_{k}}}+M\right)V+Q.$$

Extending the attention mechanism to a multi-head version enables the mechanism to consider different aspects of the label relationships. The masked multi-head attention ($MHA_{m}$) mechanism is the cascade of Equation (Equation 7), and its definition is shown as follows:(8)$$Q_{i}=XW_{i}^{Q},K_{i}=XW_{i}^{K},V_{i}=XW_{i}^{V},$$
(9)$$Z_{i}=A_{m}\left(Q_{i},K_{i},V_{i}\right),i=1,\ldots ,h,$$
(10)$$MHA_{m}\left(Q,K,V\right)=\mathrm{Concat}\left(Z_{1},Z_{2},\ldots ,Z_{h}\right)W^{O},$$

The symbols $W_{i}^{Q}\in R^{D\times d^{k}}$, $W_{i}^{K}\in R^{D\times d^{k}}$, and $W_{i}^{V}\in R^{D\times d^{v}}$ are the parameter matrices, where $d^{k}=d^{v}=D/h$. $W^{O}\in R^{d^{v}\times D}$ is the output transform matrix. *h* is the number of heads, and $Z_{i}$ is the output of each attention head.

Our masked Transformer encoder is a multi-layer architecture in which each layer consists of a masked multi-head attention mechanism and a Feed-Forward Network (FFN). With the label embedding from the previous layer $E_{l-1}$, each Transformer encoder layer exploits the label relationships and updates the label embedding $E_{l-1}$ as follows:(11)$$E_{l}^{\left(1\right)}=E_{l-1}+MHA_{m}\left({\hat{E}}_{l-1},{\hat{E}}_{l-1},E_{l-1}\right),$$
(12)$$E_{l}=E_{l}^{\left(1\right)}+FFN\left(E_{l}^{\left(1\right)}\right),$$

The symbol $\hat{\cdot }$ means the feature modified by adding position encoding. $E^{\left(1\right)}$ is the intermediate variable.

### 3.4. Final Classification and Loss Function

After being refined by the *l* Transformer encoder layer, we can obtain the final label embeddings $E_{l}\in R^{C\times D}$ and project them to the logit $y_{r}= [y_{r}^{1},y_{r}^{2},\ldots ,y_{r}^{C}]$ via a linear projection layer:(13)$$y_{r}=\sum_{c=1}^{C}\left(E_{l}^{c}W^{⊤}\right)+b,\;c=1,2,\ldots ,C,$$
where $W=\left[W_{1},W_{2},\ldots ,W_{C}\right]\in R^{C\times D}$ and $b\in R^{C}$. Then, the final label confidence $\hat{y}$ is obtained by elementwise summing up $y_{a}$ and $y_{r}$.
(14)$$\hat{y}=\sigma \left(y_{a}+y_{r}\right),$$
where σ(·) is the Sigmoid(·) function. The ground-truth label of the input image is $y= [y^{1},y^{2},\ldots ,y^{C}]$, where $y^{i}=0,1$ denotes the absence or presence of label *i* in the image. The whole framework is trained in an end-to-end manner with the traditional multi-label classification loss as follows:(15)$$L\left(y,\hat{y}\right)=\sum_{c=1}^{C}y^{c}log\left(\sigma \left(\hat{y^{c}}\right)\right)+\left(1-y^{c}\right)log\left(1-\sigma \left(\hat{y^{c}}\right)\right),$$
where σ(·) is the Sigmoid(·) function.

## 4. Experiment

### 4.1. Dataset

To verify the effectiveness of our proposed method, we compared the proposed method with the state-of-the-art on three popular multi-label remote sensing image datasets: UC-Merced multi-label [39], AID Multi-label [49], and MLRSNET [50].

#### 4.1.1. UC-Merced Multi-Label

The UC-Merced land use dataset is an aerial image dataset, which was originally built in a single-label style. The UC-Merced Multi-label dataset is a duplicate of the UC-Merced dataset and assigns all 2100 images to 17 newly defined object labels. There are 1–7 categories of objects in each image, with a size of 256 × 256. Following the division in [34,49], 80% of the image samples were exploited in the training phase, and the rest were used in the testing phase.

#### 4.1.2. AID Multi-Label

The AID Multi-label dataset is a subset of the AID dataset. This subset contains 3000 aerial images, which are assigned multiple object labels in 17 categories. AID is a large-scale aerial image dataset that contains 10,000 high-resolution aerial images collected from Google Earth imagery. All the images in AID Multi-label have a size of 600 × 600. We followed the 80-20 training–test split in [34,51].

#### 4.1.3. MLRSNet

MLRSNet is a multi-label remote sensing dataset, which contains 109,161 high-spatial-resolution optical satellite images captured from different perspectives of the world. The images in MLRSNet are annotated into 46 categories, and the number of sample images in a category varies from 1500 to 3000. The dataset covers 60 predefined categories, with one or more categories per image (up to 13). The resolution of each image ranges from 0.1 m to 10 m, and the size is fixed to 256×256. We followed the standard split used in [50]. A total of 20% of samples were randomly selected for training and 80% for testing.

### 4.2. Evaluation Metrics

In our experiment, we adopted the overall precision (OP), recall (OR) and per-category precision (CP), recall (CR), and F1-measure (CF1) for further comparison. For each image, we assigned each class as positive if its prediction probability was greater than 0.5 and then compared them with the ground-truth labels. The overall precision (OP), recall (OR) and per-category precision (CP), recall (CR), and F1-measure (CF1) are computed as follows:(16)$$\begin{array}{cccc} \; & \mathrm{OP}=\frac{\sum_{i}M_{c}^{i}}{\sum_{i}M_{p}^{i}}, & \; & \mathrm{OR}=\frac{\sum_{i}M_{c}^{i}}{\sum_{i}M_{g}^{i}},\\ \; & \mathrm{CP}=\frac{1}{C}\sum_{i}\frac{M_{c}^{i}}{M_{p}^{i}}, & \; & \mathrm{CR}=\frac{1}{C}\sum_{i}\frac{M_{c}^{i}}{M_{g}^{i}},\\ \; & \mathrm{CF}1=\frac{2\times \mathrm{CP}\times \mathrm{CR}}{\mathrm{CP}+\mathrm{CR}}, \end{array}$$
where $M_{c}^{i}$ is the number of images predicted correctly for the *i*-th category, $M_{p}^{i}$ is the number of images predicted for the *i*-th category, and $M_{g}^{i}$ is the number of ground-truth images for the *i*-th category.

Note that these results may be affected by the threshold. Among these metrics, the CF1 is more critical and comprehensive, considering both recall and precision.

### 4.3. Implementation Details

In this paper, all the experiments were conducted on a workstation with an AMD EPYC 7543 32-Core Processor, one 32 GB RAM, and a 24 GB RTX A5000 GPU. The base deep learning framework was PyTorch 1.9.0 with Python 3.8 and Cuda 11.1. Following the recent works, we adopted ResNet50 and ResNet101 [4] as our backbone, which were pre-trained on ImageNet [52] for the initialization of the parameters. The input image was first resized to 224×224 for the UC-Merced Multi-label dataset and MLRSNet dataset and 512×512 for the AID Multi-label dataset. To make a quick convergence, we followed [29], adopting the model trained on Pascal VOC 2012 [53] as the pre-trained model for the UC-Merced Multi-label dataset and the AID Multi-label dataset. RandAugment [54] and Cutout [55] were adopted for data augmentation. The size of the output feature from the backbone was $H\times W\times D^{\prime }=14\times 14\times 2048$. We reduced the channel of label embeddings *E* from $D^{\prime }=2048$ to D=512. For each layer of Transformer, we exploited 4 attention heads, and the dimensions of the attention head and feed-forward network were set to D/4=128 and 4D=2048, respectively. We did not use dropout [56] in each layer of Transformer. The batch size of each GPU was 64. Adam [57] was chosen as the optimizer to train the model for 80 epochs, with the weight decay of $10^{-2}$, $\left(\beta_{1},\beta_{2}\right)=\left(0.9,0.999\right)$, and a learning rate of $10^{-4}$.

### 4.4. Comparison with State-of-the-art Methods

#### 4.4.1. Performance on UC-Merced Multi-Label

We report the results of experiments on the UC-Merced Multi-label dataset. Our proposed method is superior to the previous methods listed in Table 1 in most of the metrics. Generally, CF1 is the primary metric since the others may be greatly affected by the chosen threshold. To be specific, S-MAT-ResNet50 improves CF1 by 9.7% compared to the baseline ResNet50 model. In comparison with MLRSSC-CNN-GNN, which is an effective GNN-based method, our S-MAT-ResNet50 achieves an improvement of 2.82% in CF1, 0.80% in CP, and 0.56% in CR. Both ResNet50-SR-Net and our method are Transformer-based methods. Our proposed MAT-ResNet50 beats ResNet50-SR-Net in the CF1, CP, CR, and OR metrics. The results show the effectiveness of our method.

#### 4.4.2. Performance on AID Multi-Label

Table 2 shows the quantitative results of the AID Multi-label dataset. The metrics are identical to the ones in the UC-Merced Multi-label dataset. As shown in Table 2, our method achieves the best CF1 score, which is more vital than other indicators. Specifically, our S-MAT-ResNet50 improves the CF1 score of ResNet50, ResNet-RBFNN, CA-ResNet-BiLSTM, AL-RN-ResNet50, MLRSSC-CNN-GNN, and ResNet50-SR-Net by 4.67%, 7.13%, 3.27%, 2.18%, 2.26%, and 0.93%, respectively. Moreover, the performance of our S-MAT-ResNet50 on CP and OP outperforms another Transformer-based approach, ResNet50-SR-Net, by 2.75% and 0.97%, respectively. These results convincingly prove the effectiveness of our proposed method.

#### 4.4.3. Performance on MLRSNet

The comparison with the previous methods on MLRSNet is reported in Table 3. We conducted experiments on two baseline CNN backbones, ResNet50 and ResNet101. As we can observe, the proposed S-MAT-CNN method outperforms the listed competitors in all metrics. It is noteworthy that, compared to the state-of-the-art method CNN-SR-Net, the proposed method improves the CF1 by 1.10% and 1.31% with ResNet50 and ResNet101 as the backbone, respectively.

## 5. Ablation Studies

In this section, we perform exhaustive experiments to analyze the essential components of our proposed method. Firstly, we analyzed the contribution of each component in our S-MAT framework. Secondly, the effects of different settings of the Transformer encoder were analyzed, including the number of attention heads, the number of encoder layers, and different position encoding. Thirdly, we discuss the settings of the generation and application of the mask in masked attention, including the selection of *k* and the position to apply masked attention. For simplicity, in this section, we shall abbreviate “UC-Merced Multi-label” and “AID Multi-label” to “UC-Merced” and “AID” in all the tables, respectively.

### 5.1. Contributions of the Proposed Method

To investigate the effectiveness of each component in our proposed method, we gradually removed SDM and MAT from the complete framework. Meanwhile, to comprehensively explore the contribution of the SDM, we studied the effect of $y_{a}$. We first removed MAT to evaluate the SDM by feeding the raw label embeddings output from the SDM into the final linear projection layer. Then, we replaced the SDM with an MLP layer to evaluate the contribution of the SDM. The result is shown in Table 4. On three popular benchmark datasets, the SDM and MAT significantly promote the baseline, respectively, which convincingly proves the effectiveness of the SDM and MAT. By observing the first three rows in the table, it can be found that the promotions of the SDM are from two aspects. On the one hand, the high-level features from the backbone are disentangled into more discriminative label embedding features. On the other, when the prediction $y_{a}$ takes effect, the constraints from the loss interpreted in Equation (Equation 15) enable the SDM to generate more accurate class-specific activation map. Moreover, we found that the combination of the SDM and MAT maximizes the performance of CF1 (+9.70% on UC-Merced, +4.67% on AID, +6.96% on MLRSNet), which demonstrates that the SDM and MAT are complementary.

### 5.2. Number of Attention Heads

We demonstrated the effectiveness of each component in our proposed approach. In this part, we explore the performance curve of our proposed method by varying the number of attention heads. To avoid the effect of masked attention, we exploited standard dot-product attention instead of masked attention in this part. We set the number of layers in Transformer L=1 and gradually increased the number of attention heads *H* from 1 to 12. The results are shown in Figure 4. The performance curves across UC-Merced Multi-label and AID Multi-label are similar. Appropriately increasing the *H* of multi-head attention enhances the performance of CF1. However, when there are too many attention heads, the performance drops since not enough features are assigned to each attention head. When the number of attention heads H=4, the performance of CF1 reaches its summit.

### 5.3. Number of Layers in Transformer

To evaluate the effect of the different number of layers *L* in Transformer on CF1, we changed *L* from 1 to 6. To avoid the effect of masked attention, we exploited standard dot-product attention instead of masked attention in this part. Following the best setting in the last part, the number of attention head *H* was set to 4. In Figure 5, we show the result of varying the number of layers in Transformer. The results show that the performance of CF1 ascends to the peak when three layers are stacked. By stacking more layers, the performance drops continually on both the UC-Merced and AID datasets.

### 5.4. Position to Apply Masked Attention

In this part, we attempt to figure out the best position to apply our masked attention by replacing the standard attention with our masked attention in different layers. As shown in Table 5, we applied masked attention in (1) the first layer, (2) the second layer, (3) the last layer, and (4) all layers, respectively. The results show that, compared to standard dot-product attention, the application of masked attention in any layer improves the performance of CF1. Moreover, when the masked attention is applied in all layers, we can obtain the best performance. Compared to the model without masked attention (Line 0), our proposed masked attention provides +1.38%, +0.95%, and +1.92%CF1 on UC-Merced, AID, and MLRSNet, respectively, without introducing any extra parameters.

### 5.5. Selection of *k* in the Generation of the Mask

The selection of *k* is a fundamental setting in the generation of the mask. With an increasing *k*, more label dependencies computed in the self-attention operator are filtered out. The results are shown in Figure 6. Note that, if k=0, no label dependency is available and the model will not converge. Thus, we do not show any result for k=0 in Figure 6. In this part, the number of layers in MAT *L* was set to 1. When *k* was reducing from 100 to 25, our model obtained an increasing CF1. However, when *k* was less than 25, the CF1 dropped since some categories with high confidence would be ignored as well.

### 5.6. Position Embedding

Since position embedding retains the relative position of the input sequence in the original Transformer, we attempted to figure out its effectiveness in our method. As shown in Table 6, when the label embeddings are elementwise summed up with the position embedding, the performance of CF1 improves around 0.1% on MLRSNet. Meanwhile, position embedding barely contributes to the performance of UC-Merced and AID. We speculate that since there are more categories and more complicated label relationships in MLRSNet, the model with position embedding is slightly better than the one without it.

## 6. Visualization

In this section, qualitative results are presented to further reveal the effectiveness of the proposed models. Specifically, we firstly visualize the class-specific activation map in the SDM to show if the SDM could provide precise localization. Then, we visualize the learned inter-class relationship matrix in MAT to demonstrate the ability to capture the co-existence of objects.

### 6.1. Visualization of Class-Specific Activation Map

To investigate the capability of locating the position of each category in the image, we visualize the class-specific activation map *M* in the SDM. Some qualitative results are shown in Figure 7. Each row presents the raw image and the corresponding class-specific activation map of each ground-truth label. We can find that the SDM is able to locate the corresponding region of each category in the image. For instance, the ground-truth labels of the raw image in the third row are “building”, “ship”, “tree”, and “pavement”. Our SDM module can provide accurate localization of these four categories. In addition, the accurate activation map leads to more precise label embeddings for modeling the inter-class dependencies in MAT.

### 6.2. Visualization of Relationship Matrix in MAT

To further demonstrate the effectiveness of our method, we visualize the relationship matrix in the last layer of MAT. Partial results of the visualization are shown in Figure 8. Each row in Figure 8 consists of a raw image and the visualization of the corresponding relation matrix in MAT. We can observe that our proposed MAT is able to explore and capture the accurate label dependencies. Specifically, for the input image in the first row, the ground-truth labels are (“airplane”, “bare soil”, “buildings”, “cars”, “grass”, “pavement”, “trees”). In the relation matrix on the right, we can find that A(trees, airplane), A(pavement, airplane), A(grass, airplane), A(buildings, airplane), and A(bare soil, airplane) are ranked top in the column of “airplane”. This indicates that the correlations between the label pair (“trees”, “airplane”), (“pavement”, “airplane”), (“grass”, “airplane”), (“buildings”, “airplane”), and (“bare soil”, “airplane”) are higher than the others (greater than 0.7). Meanwhile, for the input image in the second row, the ground-truth labels are (“bare soil”, “buildings”, “cars”, “courts”, “grass”, “pavement”, “trees”). In the relation matrix on the right, we can find that A(trees, courts), A(pavement, courts), A(grass, courts), A(cars, courts), A(buildings, courts), and A(bare soil, courts) are ranked top in the column of “courts”. This indicates that the correlations between the label pair (“trees”, “courts”), (“pavement”, “courts”), (“grass”, “courts”), (“cars”, “courts”), (“buildings”, “courts”), and (“bare soil”, “courts”) are higher than the others (greater than 0.7). These convincingly demonstrate that our proposed method can explore, capture, and exploit the label relationships in specific images.

## 7. Discussion

In this section, we demonstrate the analyses of the experimental results and differences between our S-MAT and previous methods. As shown in Table 1 and Table 2, we can find that S-MAT not only improves the performance of precision and recall, but also keeps them in balance. Consequently, S-MAT always wins the comparison of CF1, which considers both precision and recall. Secondly, the RNN-based methods, such as CA-ResNet-BiLSTM and CM-GM-N-R-BiLSTM, barely surpass the baseline ResNet50. Limited by modeling as the pre-set sequence, RNN-based methods are incapable of capturing the accurate inter-class relationship. Different from the RNN, Transformer can compute the attention matrix among the labels. Therefore, Transformer-based methods, such as ResNet50-SR-Net and our S-MAT-ResNet50, are the superior approaches to model the relationships between pairwise labels. Moreover, compared with SR-NET, in the S-MAT framework, MAT replaces standard dot-product attention with masked attention. The latter can filter out redundancy to obtain a more accurate inter-class relationship. Meanwhile, the SDM disentangles the global feature map to generate the label embeddings and, thus, introduce valuable spatial information for feature representation. As demonstrated in Table 4, the combination of these two modules maximizes recognition performance on the three benchmark datasets.

Figure 9 shows a qualitative example on the AID Multi-label dataset. The first column is the input image. The second column and the third one are the methods and the output predictions, respectively. Green labels denote true positive; red labels denote false positive; gray ones denote false negative. We can find that the baseline ResNet-50 failed to distinguish the objects with similar appearances, such as “trees”, “grass”, and “field” since the model only utilizes the global spatial features. The other methods obtain a lower false positive rate than the baseline by dealing with label dependencies. Those inconspicuous objects, such as “cars” and “dock”, can not be recognized by ResNet-50 and CA-ResNet-BiLSTM. While SR-Net misses the prediction of “dock”, S-MAT can recognize both “cars” and “dock” accurately and obtain a better performance.

It is worth noting that our model leans upon massive annotated large-scale datasets to learn the semantic context and the label dependencies of a visual scene. However, massive annotated data are costly and rare, while most aerial scene images are unlabeled. Therefore, learning the visual feature via self-supervised learning is our research topic in the future.

## 8. Conclusions

Multi-label aerial scene image classification is a fundamental task in the computer vision area. Label dependency is a vital factor for multi-label learning. In this paper, we aimed to model a more accurate inter-class relationship by removing the redundant label dependencies among the low confidence categories. We presented a Transformer-based framework S-MAT for multi-label aerial scene image classification. Specifically, we proposed two plug-and-play modules, the Semantic Disentanglement Module (SDM) and Masked-Attention Transformer (MAT). The SDM aims to locate the semantic region of each category and conduct semantic disentanglement to generate label embeddings for MAT. MAT not only captures and models the complicated inter-class relationships in a specific image, but filters out the redundant label dependencies by replacing the standard dot-product attention in the standard Transformer architecture with the proposed masked attention. Especially, the masked attention significantly improves the performance without introducing additional parameters. Therefore, our S-MAT consistently outperforms the prior works on three widely used and challenging remote sensing image datasets including UC-Merced Multi-label, AID Multi-label, and MLRSNet. In addition, quantitative and qualitative ablation studies and visualizations convincingly proved the effectiveness of the essential components of our method under different factors. In the future, we will focus on self-supervised learning for multi-label aerial scene image classification.

## Figures and Tables

**Figure 1 sensors-22-05433-f001:**
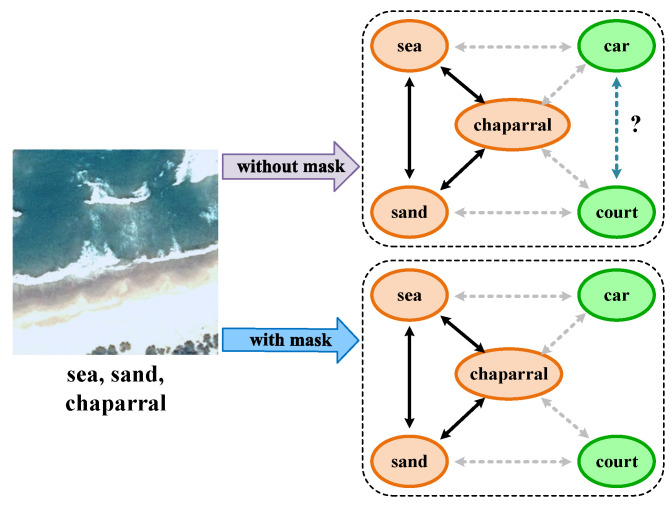
Illustration of the effect of mask attention.

**Figure 2 sensors-22-05433-f002:**
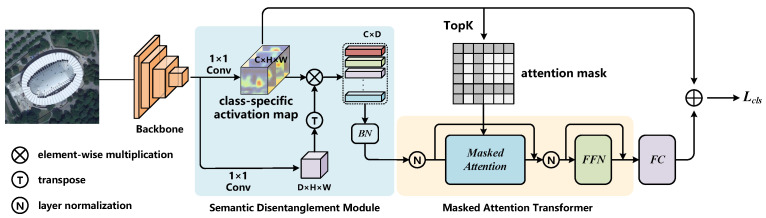
The overall framework of our proposed method.

**Figure 3 sensors-22-05433-f003:**
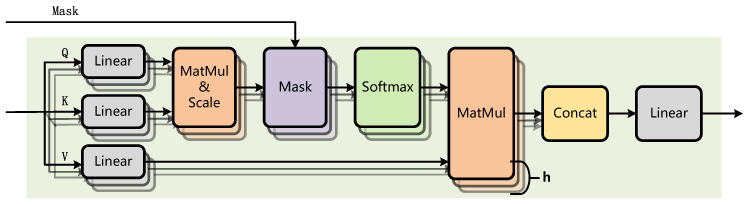
Illustration of the masked attention.

**Figure 4 sensors-22-05433-f004:**
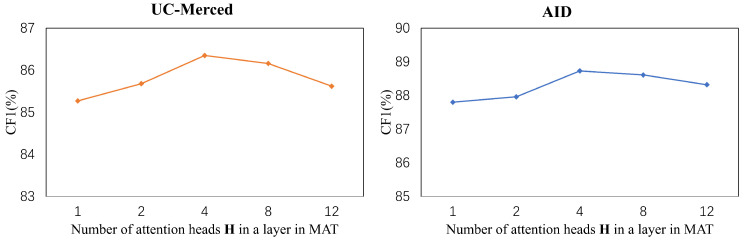
The results of the ablation studies on the number of attention heads in a layer in MAT.

**Figure 5 sensors-22-05433-f005:**
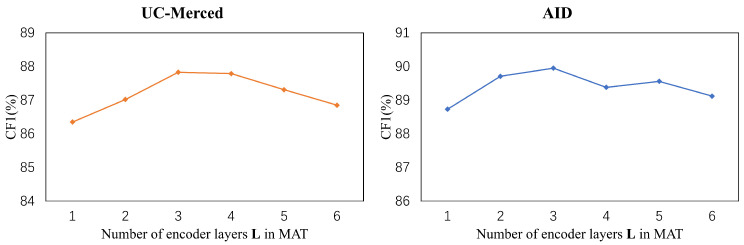
The results of ablation studies on the number of encoder layers in MAT.

**Figure 6 sensors-22-05433-f006:**
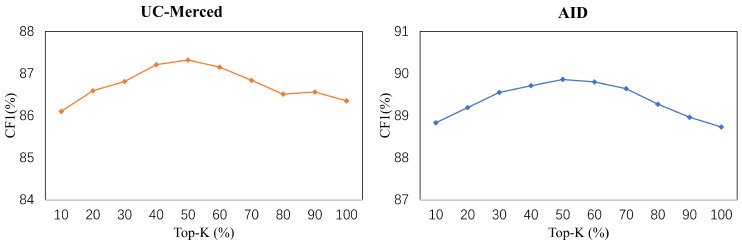
The results of ablation studies on the selection of *k* in the generation of the mask in MAT.

**Figure 7 sensors-22-05433-f007:**
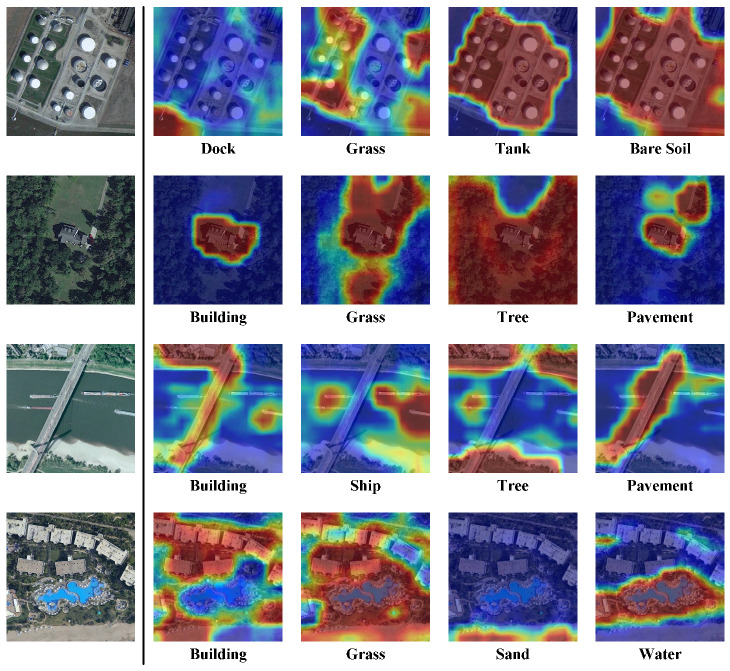
Visualization of the class-specific activation map in the SDM on the AID Multi-label dataset.

**Figure 8 sensors-22-05433-f008:**
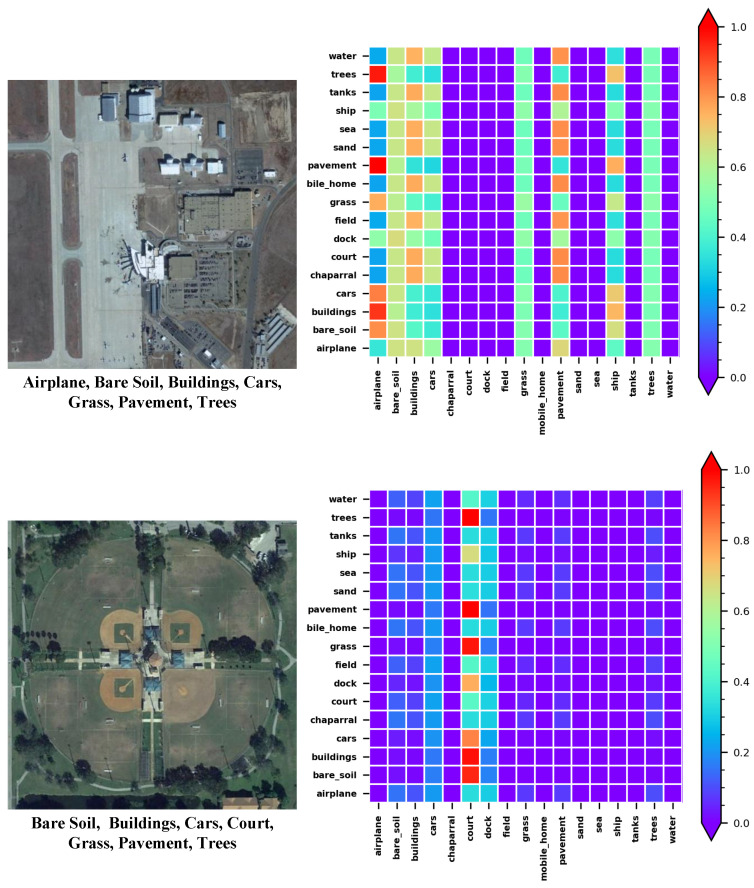
Visualization of the relation matrix in MAT on the AID Multi-label dataset.

**Figure 9 sensors-22-05433-f009:**
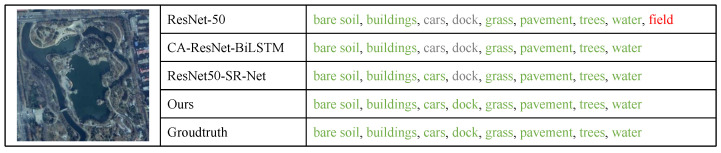
Qualitative results on the AID Multi-label dataset.

**Table 1 sensors-22-05433-t001:** Comparisons of our method with previous state-of-the-art methods on the UC-Merced Multi-label dataset. Among all the metrics, CF1 is the primary metric. The bold means the best performance. All metrics are in %.

Method	CF1	CP	CR	OP	OR
ResNet50 [4]	79.51	88.52	78.91	80.70	81.97
ResNet-RBFNN [58]	80.58	86.21	83.72	79.92	84.59
CA-ResNet-BiLSTM [34]	81.47	86.12	84.26	77.94	89.02
CM-GM-N-R-BiLSTM [43]	81.58	88.57	85.20	81.60	89.65
AL-RN-ResNet50 [49]	86.76	**88.81**	87.07	86.12	84.26
MLRSSC-CNN-GNN [42]	86.39	87.11	88.41	-	-
ResNet50-SR-Net [29]	88.67	87.96	89.40	**93.52**	91.51
**S-MAT-ResNet50 (Ours)**	**89.21**	87.97	**89.96**	92.94	**92.38**

**Table 2 sensors-22-05433-t002:** Comparisons of our method with previous state-of-the-art methods on the AID Multi-label dataset. Among all the metrics, CF1 is the primary metric. The bold means the best performance. All metrics are in %.

Method	CF1	CP	CR	OP	OR
ResNet50 [4]	86.23	89.31	85.65	72.39	52.82
ResNet-RBFNN [58]	83.77	82.84	88.32	60.85	70.45
CA-ResNet-BiLSTM [34]	87.63	89.03	88.95	79.50	65.60
AL-RN-ResNet50 [49]	88.72	91.00	88.95	80.81	71.12
MLRSSC-CNN-GNN [42]	88.64	89.83	90.20	-	-
ResNet50-SR-Net [29]	89.97	89.42	**90.52**	87.24	**82.25**
**S-MAT-ResNet50 (Ours)**	**90.90**	**92.17**	89.69	**88.21**	80.70

**Table 3 sensors-22-05433-t003:** Comparisons of our method with previous state-of-the-art methods on the MLRSNet dataset. ⋆ denotes our implementation. Among all the metrics, CF1 is the primary metric. The bold means the best performance. All metrics are in %.

Method	CF1	CP	CR	OP	OR
ResNet50 [4]	75.30	-	-	-	-
ResNet50 ${}^{⋆}$ [4]	81.35	80.85	81.56	82.19	82.70
ResNet50-SR-Net [29]	87.21	87.08	87.34	88.79	86.73
**S-MAT-ResNet50 (Ours)**	**88.31**	**87.80**	**88.79**	**90.93**	**91.02**
ResNet101 [4]	76.18	-	-	-	-
ResNet101 ${}^{⋆}$ [4]	81.89	81.42	82.03	82.65	82.89
ResNet101-SR-Net [29]	87.55	87.84	87.26	89.41	87.48
**S-MAT-ResNet101 (Ours)**	**88.86**	**88.67**	**88.93**	**91.21**	**91.44**

**Table 4 sensors-22-05433-t004:** Ablation studies on the essential components of our proposed method. The symbol **✓** represents the component in this column is in use. The red font denotes the improvement over the baseline. The bold means the best performance.

Component		Prediction	CF1		
**SDM**	**MAT**	$y_{a}$	**UC-Merced**	**AID**	**MLRSNet**
-	-	-	79.51	86.23	81.35
**✓**	-	-	82.35 ↑2.84	87.16 ↑0.93	83.46 ↑2.11
**✓**	-	**✓**	84.18 ↑4.67	87.78 ↑1.55	84.71 ↑3.36
-	**✓**	-	85.61 ↑6.10	88.26 ↑2.03	85.93 ↑4.58
**✓**	**✓**	-	87.59 ↑8.08	90.14 ↑3.91	86.48 ↑5.13
**✓**	**✓**	**✓**	**89.21** ↑9.70	**90.90** ↑4.67	**88.31** ↑6.96

**Table 5 sensors-22-05433-t005:** Ablation studies on the position to apply masked attention. The symbol **✓** represents that the masked attention is in use in this layer. The bold means the best performance.

Masked Attention			CF1		
**Layer 1**	**Layer 2**	**Layer 3**	**UC-Merced**	**AID**	**MLRSNet**
-	-	-	87.83	89.95	86.39
**✓**	-	-	88.57	90.26	87.42
-	**✓**	-	88.12	90.08	86.81
-	-	**✓**	88.85	90.43	87.70
**✓**	**✓**	**✓**	**89.21**	**90.90**	**88.31**

**Table 6 sensors-22-05433-t006:** Ablation studies on position embedding. The symbol **✓** represents that position embedding is in use. The bold means the best performance.

Method	CF1		
**Position Embedding**	**UC-Merced**	**AID**	**MLRSNet**
-	89.19	90.89	88.22
**✓**	**89.21**	**90.90**	**88.31**

## Data Availability

Not applicable.

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
