# Peer review of "S-MAT: Semantic-Driven Masked Attention Transformer for Multi-Label Aerial Image Classification"

_sensors, 2022, doi:10.3390/s22145433_

Round 1
Reviewer 1 Report
SUMMARY
The article submitted for review is devoted to a topical issue: "A Semantic-Driven Masked-Attention Transformer for Multi-Label Aerial Image Classification". The research problem is that the classification of aerial photographs with multiple labels is a complex study and a longstanding problem. The problem exists in the field of remote sensing, and thus the topic represents a certain scientific and applied deficit.
The authors carried out interesting studies that are original and have scientific novelty and practical significance. With the obvious advantages of the article, it has disadvantages, which will be discussed below.
COMMENTS
1. I would like to see in the Abstract a more clearly formulated scientific problem and an applied problem, then it will become more clear what this article is about and the Abstract will fulfill its role more efficiently.
2. In addition, in the Abstract I would like to see some quantitative expression of the result achieved. At the moment, it is a text that is difficult to perceive and, as a result, the abstract does not fully reflect the content of the article. For a topic such as a Semantic-driven masked-attention transformer for multi-label aerial image classification, it is necessary to add some numerical expressions for a clearer understanding of scientific novelty and practical significance.
3. Perhaps the keywords should be supplemented with more specific terms in order to move away from the general to the particular, since the article is devoted to a specific issue, not a direction.
4. In the Introduction section, I would like to see a slightly larger number of analyzed sources. The topic of the article is relevant, modern, and it would be necessary to increase the number of analyzed sources from 19 to 25–30, then it will be possible to speak more fully about the scientific novelty of the study.
5. The quality of the image in Figure 1 on the left side should be presented in a more correct form. At the moment it looks a little blurry.
6. Also noteworthy is the not entirely methodologically correct representation of Figure 1. For example, it has a specific title, but the description and interpretation of this figure are also given in the caption, it is necessary to clearly separate the name of the figure and its textual interpretation.
7. Probably, the statement that “for some categories with low confidence, interclass relationships cannot be effectively estimated” needs some additional interpretation. This is about 1 paragraph after Figure 1. A smoother transition from section 1 to section 2 should be provided in order for the text to be perceived more smoothly.
8. In addition, it is necessary to present a more clearly formulated goal, research objectives, the scientific problem being solved and scientific novelty.
9. Subparagraph 2.2.3 ends with a figure, the textual description of which is similar to the remark above. In addition, it is methodologically not entirely correct to end the section with a drawing; a more extended textual interpretation should be given after it.
10. Obviously, the authors sent an unfinished version of the article, since subsection 2.3 is missing, you should either delete the title of this section or insert this section, while it is important that the semantic load of the text is preserved. This issue should be considered more seriously by the authors.
11. Formulas 1, 2 and 3 need some additional interpretation on the applied values.
12. The same remark can be applied to formulas 4 and 5.
13. The experiment in section 4 is described in the format of the protocol, which makes the article quite difficult to understand, perhaps it should be diluted with a more detailed textual interpretation of the ongoing research.
14. Figure 4 is a set of a large number of graphs, consisting of one curve on each. This makes such a graphical representation difficult to perceive for the viewer, the format of the graphs should probably be reconsidered, but this is at the discretion of the authors.
15. The article should be supplemented with a detailed comparison of the results obtained by the authors with the results obtained earlier by other authors. This will allow us to talk about the scientific novelty and scientific significance of the study. In traditional article formats, such a section is called Discussion, but in the presented article it is not separated into a separate section. This issue should be taken seriously.
16. The “Conclusions” section needs a slightly more detailed interpretation of the result obtained, a more specific presentation of the main scientific achievements of the authors, a demonstration of scientific novelty, and a possibly more detailed description of the prospects and possible directions for the development of the study.
Author Response
The response is uploaded as a PDF file. Please see the attachment.

Reviewer 2 Report
Title of the paper: "S-MAT: Semantic-Driven Masked-Attention Transformer for Multi-Label Aerial Image Classification"
As a researcher working in the same field, I am impressed by the technique introduced in the paper, because it sheds new light on the earlier results of several authors and obviously can be successfully used in practice. From this point of view, the subject of the paper fits well with the scope of the journal (Sensors).
The paper is ended with numerical simulations that corroborate the theoretical results.
This manuscript contains new ideas and good results that help other researchers.
The decision is too major revision for publication in the "Sensors".
Therefore, I recommend publishing this work after taking these points into account.
1-The English writing of the paper is required to be improved. Please check the manuscript carefully for typos and grammatical errors. I found some typos and grammatical errors within this manuscript, which have been excluded from my review. In addition, the English structure of the article, including punctuation, semicolon, and other structures, must be carefully reviewed.
2-In the introduction, the authors did not provide a strong motivation for the paper and the obtained results. In addition, they should discuss the main contributions of their work in detail after the motivation part. Then they should summarize the main structure of their paper in brief at the end of the introduction.
3-The literature review about the problem under study is not adequate. I suggest the authors keep up-to-date the introductory part with the recent relevant developments and publications.
4-I found no comparative results within this manuscript. Some comparative results with the other methods available in the literature would be expected in the revision.
5-The introduction needs to be improved by the recent developments in the field of numerical simulation and stability as well as its applications. For this purpose, the authors can add the following references to enrich the introductory section:
*A numerical method for solving the nonlinear equations of Emden-Fowler models, Journal of Ocean Engineering and Science, 2022. doi.org/10.1016/j.joes.2022.04.019.
6-Future recommendations should be added to assist other researchers to extend the presented research analysis.
Author Response

(The authors gave the same response as above.)

Reviewer 3 Report
In this paper, the author attempts to construct a Semantic-driven MaskedAttention Transformer through Masked Attention Transformer and Semantic Disentanglement Module to solve the problem of redundant labels in multi-label aerial image classification. From the results, the extraction effect is improved. At the same time, there are some problems that need to be solved.
1. In the introduction, it is mentioned that “land cover objects have different scales”. There is no corresponding explanation for the different scales, does it mean resolution or feature map? At the same time, there should be some content in the paper to describe this part.
2. In Section 3.1, it is mentioned that “ The standard dot-product attention with residual path is defined in equation 1 ”, The dot-product attention is not equation 1.
3. In Section 4.1.3, the proportion of training samples and test samples is not the same as the other two datasets. In addition, does the model converge stably under such a setting?
4. In Fig. 6, the image on the second row should have an appropriate description and explanation.
5. In Section 3.3, it is mentioned that “ retain some redundancy in the generation of masks by adjusting the proportion to be filtered out ”. How is this proportion adjusted?
Round 2
Reviewer 2 Report
No comments.
The paper has now been excellent.
In my opinion, the paper is well organized and the results of the paper are correct, interesting and I strongly recommend its publication.